# Action Centered Contextual Bandits

**Kristjan Greenewald**
Department of Statistics
Harvard University
kgreenewald@fas.harvard.edu

**Ambuj Tewari**
Department of Statistics
University of Michigan
tewaria@umich.edu

**Predrag Klasnja**
School of Information
University of Michigan
klasnja@umich.edu

**Susan Murphy**
Departments of Statistics and Computer Science
Harvard University
samurphy@fas.harvard.edu

## Abstract

Contextual bandits have become popular as they offer a middle ground between very simple approaches based on multi-armed bandits and very complex approaches using the full power of reinforcement learning. They have demonstrated success in web applications and have a rich body of associated theoretical guarantees. Linear models are well understood theoretically and preferred by practitioners because they are not only easily interpretable but also simple to implement and debug. Furthermore, if the linear model is true, we get very strong performance guarantees. Unfortunately, in emerging applications in mobile health, the time-invariant linear model assumption is untenable. We provide an extension of the linear model for contextual bandits that has two parts: baseline reward and treatment effect. We allow the former to be complex but keep the latter simple. We argue that this model is plausible for mobile health applications. At the same time, it leads to algorithms with strong performance guarantees as in the linear model setting, while still allowing for complex nonlinear baseline modeling. Our theory is supported by experiments on data gathered in a recently concluded mobile health study.

## 1 Introduction

In the theory of sequential decision-making, contextual bandit problems (Tewari & Murphy, 2017) occupy a middle ground between multi-armed bandit problems (Bubeck & Cesa-Bianchi, 2012) and full-blown reinforcement learning (usually modeled using Markov decision processes along with discounted or average reward optimality criteria (Sutton & Barto, 1998; Puterman, 2005)). Unlike bandit algorithms, which cannot use any side-information or context, contextual bandit algorithms can learn to map the context into appropriate actions. However, contextual bandits do not consider the impact of actions on the evolution of future contexts. Nevertheless, in many practical domains where the impact of the learner's action on future contexts is limited, contextual bandit algorithms have shown great promise. Examples include web advertising (Abe & Nakamura, 1999) and news article selection on web portals (Li et al., 2010).

An influential thread within the contextual bandit literature models the expected reward for any action in a given context using a linear mapping from a $d$-dimensional context vector to a real-valued *reward*. Algorithms using this assumption include LinUCB and Thompson Sampling, for both of which regret bounds have been derived. These analyses often allow the context sequence to be chosen adversarially, but require the linear model, which links rewards to contexts, to be time-invariant. There has been little effort to extend these algorithms and analyses when the data follow an unknown nonlinear or time-varying model.

In this paper, we consider a particular type of non-stationarity and non-linearity that is motivated by problems arising in mobile health (mHealth). Mobile health is a fast developing field that uses mobile and wearable devices for health care delivery. These devices provide us with a real-time stream of dynamically evolving contextual information about the user (location, calendar, weather, physical activity, internet activity, etc.). Contextual bandit algorithms can learn to map this contextual information to a set of available intervention options (e.g., whether or not to send a medication reminder). However, human behavior is hard to model using stationary, linear models. We make a fundamental assumption in this paper that is quite plausible in the mHealth setting. In these settings, there is almost always a "do nothing" action usually called action 0. The expected reward for this action is the *baseline reward* and it can change in a very non-stationary, non-linear fashion. However, the *treatment effect* of a non-zero action, i.e., the incremental change over the baseline reward due to the action, can often be plausibly modeled using standard stationary, linear models.

We show, both theoretically and empirically, that the performance of an appropriately designed action-centered contextual bandit algorithm is agnostic to the high model complexity of the baseline reward. Instead, we get the same level of performance as expected in a stationary, linear model setting. Note that it might be tempting to make the entire model non-linear and non-stationary. However, the sample complexity of learning very general non-stationary, non-linear models is likely to be so high that they will not be useful in mHealth where data is often noisy, missing, or collected only over a few hundred decision points.

We connect our algorithm design and theoretical analysis to the real world of mHealth by using data from a pilot study of HeartSteps, an Android-based walking intervention. HeartSteps encourages walking by sending individuals contextually-tailored suggestions to be active. Such suggestions can be sent up to five times a day–in the morning, at lunchtime, mid-afternoon, at the end of the workday, and in the evening–and each suggestion is tailored to the user's current context: location, time of day, day of the week, and weather. HeartSteps contains two types of suggestions: suggestions to go for a walk, and suggestions to simply move around in order to disrupt prolonged sitting. While the initial pilot study of HeartSteps micro-randomized the delivery of activity suggestions (Klasnja et al., 2015; Liao et al., 2015), delivery of activity suggestions is an excellent candidate for the use of contextual bandits, as the effect of delivering (vs. not) a suggestion at any given time is likely to be strongly influenced by the user's current context, including location, time of day, and weather.

This paper's main contributions can be summarized as follows. We introduce a variant of the standard linear contextual bandit model that allows the baseline reward model to be quite complex while keeping the treatment effect model simple. We then introduce the idea of using action centering in contextual bandits as a way to decouple the estimation of the above two parts of the model. We show that action centering is effective in dealing with time-varying and non-linear behavior in our model, leading to regret bounds that scale as nicely as previous bounds for linear contextual bandits. Finally, we use data gathered in the recently conducted HeartSteps study to validate our model and theory.

## 1.1 Related Work

Contextual bandits have been the focus of considerable interest in recent years. Chu et al. (2011) and Agrawal & Goyal (2013) have examined UCB and Thompson sampling methods respectively for linear contextual bandits. Works such as Seldin et al. (2011), Dudik et al. (2011) considered contextual bandits with fixed policy classes. Methods for reducing the regret under complex reward functions include the nonparametric approach of May et al. (2012), the "contextual zooming" approach of Slivkins (2014), the kernel-based method of Valko et al. (2013), and the sparse method of Bastani & Bayati (2015). Each of these approaches has regret that scales with the complexity of the overall reward model including the baseline, and requires the reward function to remain constant over time.

## 2 Model and Problem Setting

Consider a contextual bandit with a baseline (zero) action and $N$ non-baseline arms (actions or treatments). At each time $t = 1, 2, \ldots$, a context vector $\bar{s}_t \in \mathbb{R}^{d'}$ is observed, an action $a_t \in \{0, \ldots, N\}$ is chosen, and a reward $r_t(a_t)$ is observed. The bandit learns a mapping from a state vector $s_{t,a_t}$ depending on $\bar{s}_t$ and $a_t$ to the expected reward $r_t(s_{t,a_t})$. The state vector $s_{t,a_t} \in \mathbb{R}^d$ is a function of $a_t$ and $\bar{s}_t$. This form is used to achieve maximum generality, as it allows for infinite possible actions so long as the reward can be modeled using a $d$-dimensional $s_{t,a}$. In the most

unstructured case with $N$ actions, we can simply encode the reward with a $d = Nd'$ dimensional $s_{t,a_t}^T = [I(a_t = 1)\bar{s}_t^T, \dots, I(a_t = N)\bar{s}_t^T]$ where $I(\cdot)$ is the indicator function.

For maximum generality, we assume the context vectors are chosen by an adversary on the basis of the history $\mathcal{H}_{t-1}$ of arms $a_\tau$ played, states $\bar{s}_\tau$, and rewards $r_\tau(\bar{s}_\tau, a_\tau)$ received up to time $t-1$, i.e.,

$$\mathcal{H}_{t-1} = \{a_\tau, \bar{s}_t, r_\tau(\bar{s}_\tau, a_\tau), i = 1, \dots, N, \tau = 1, \dots, t-1\}.$$

Consider the model $E[r_t(\bar{s}_t, a_t)|\bar{s}_t, a_t] = \bar{f}_t(\bar{s}_t, a_t)$, where $\bar{f}_t$ can be decomposed into a fixed component dependent on action and a time-varying component that does not depend on action:

$$E[r_t(\bar{s}_t, a_t)|\bar{s}_t, a_t] = \bar{f}_t(\bar{s}_t, a_t) = f(s_{t,a_t})I(a_t > 0) + g_t(\bar{s}_t),$$

where $\bar{f}_t(\bar{s}_t, 0) = g_t(\bar{s}_t)$ due to the indicator function $I(a_t > 0)$. Note that the optimal action depends in no way on $g_t$, which merely confounds the observation of regret. We hypothesize that the regret bounds for such a contextual bandit asymptotically depend only on the complexity of $f$, not of $g_t$. We emphasize that we do not require any assumptions about or bounds on the complexity or smoothness of $g_t$, allowing $g_t$ to be arbitrarily nonlinear and to change abruptly in time. These conditions create a partially agnostic setting where we have a simple model for the interaction but the baseline cannot be modeled with a simple linear function. In what follows, for simplicity of notation we drop $\bar{s}_t$ from the argument for $r_t$, writing $r_t(a_t)$ with the dependence on $\bar{s}_t$ understood.

In this paper, we consider the linear model for the reward difference at time $t$:

$$r_t(a_t) - r_t(0) = f(s_{t,a_t})I(a_t > 0) + n_t = s_{t,a_t}^T \theta I(a_t > 0) + n_t \tag{1}$$

where $n_t$ is zero-mean sub-Gaussian noise with variance $\sigma^2$ and $\theta \in \mathbb{R}^d$ is a vector of coefficients. The goal of the contextual bandit is to estimate $\theta$ at every time $t$ and use the estimate to decide which actions to take under a series of observed contexts. As is common in the literature, we assume that both the baseline and interaction rewards are bounded by a constant for all $t$.

The task of the action-centered contextual bandit is to choose the probabilities $\pi(a, t)$ of playing each arm $a_t$ at time $t$ so as to maximize expected differential reward

$$\mathbb{E}[r_t(a_t) - r_t(0)|\mathcal{H}_{t-1}, s_{t,a}] = \sum_{a=0}^{N} \pi(a, t)\mathbb{E}[r_t(a) - r_t(0)|\mathcal{H}_{t-1}, s_{t,a}] \tag{2}$$
$$= \sum_{a=0}^{N} \pi(a, t)s_{t,a}^T \theta I(a > 0).$$

This task is closely related to obtaining a good estimate of the reward function coefficients $\theta$.

## 2.1 Probability-constrained optimal policy

In the mHealth setting, a contextual bandit must choose at each time point whether to deliver to the user a behavior-change intervention, and if so, what type of intervention to deliver. Whether or not an intervention, such as an activity suggestion or a medication reminder, is sent is a critical aspect of the user experience. If a bandit sends too few interventions to a user, it risks the user's disengaging with the system, and if it sends too many, it risks the user's becoming overwhelmed or desensitized to the system's prompts. Furthermore, standard contextual bandits will eventually converge to a policy that maps most states to a near-100% chance of sending or not sending an intervention. Such regularity could not only worsen the user's experience, but ignores the fact that users have changing routines and cannot be perfectly modeled. We are thus motivated to introduce a constraint on the size of the probabilities of delivering an intervention. We constrain $0 < \pi_{\min} \le 1 - \mathbb{P}(a_t = 0|\bar{s}_t) \le \pi_{\max} < 1$, where $\mathbb{P}(a_t = 0|\bar{s}_t)$ is the conditional bandit-chosen probability of delivering an intervention at time $t$. The constants $\pi_{\min}$ and $\pi_{\max}$ are not learned by the algorithm, but chosen using domain science, and might vary for different components of the same mHealth system. We constrain $\mathbb{P}(a_t = 0|\bar{s}_t)$, not each $\mathbb{P}(a_t = i|\bar{s}_t)$, as which intervention is delivered is less critical to the user experience than being prompted with an intervention in the first place. User habituation can be mitigated by implementing the nonzero actions ($a = 1, \dots, N$) to correspond to several *types* or *categories* of messages, with the exact message sent being randomized from a set of differently worded messages.

Conceptually, we can view the bandit as pulling two arms at each time $t$: the probability of sending a message (constrained to lie in $[\pi_{\min}, \pi_{\max}]$) and which message to send if one is sent. While these probability constraints are motivated by domain science, these constraints also enable our

proposed action-centering algorithm to effectively orthogonalize the baseline and interaction term rewards, achieving sublinear regret in complex scenarios that often occur in mobile health and other applications and for which existing approaches have large regret.

Under this probability constraint, we can now derive the optimal policy with which to compare the bandit. The policy that maximizes the expected reward (2) will play the optimal action

$$a_t^* = \arg \max_{i \in \{0,...,N\}} s_{t,i}^T \theta I(i > 0),$$

with the highest allowed probability. The remainder of the probability is assigned as follows. If the optimal action is nonzero, the optimal policy will then play the zero action with the remaining probability (which is the minimum allowed probability of playing the zero action). If the optimal action is zero, the optimal policy will play the nonzero action with the highest expected reward

$$\bar{a}_t^* = \arg \max_{i \in \{1,...,N\}} s_{t,i}^T \theta$$

with the remaining probability, i.e. $\pi_{\min}$. To summarize, under the constraint $1 - \pi_t^*(0,t) \in [\pi_{\min}, \pi_{\max}]$, the expected reward maximizing policy plays arm $a_t$ with probability $\pi^*(a,t)$, where

$$\text{If } a_t^* \neq 0: \pi^*(a_t^*, t) = \pi_{\max}, \quad \pi^*(0,t) = 1 - \pi_{\max}, \quad \pi^*(a,t) = 0 \,\forall a \neq 0, a_t^* \qquad (3)$$
$$\text{If } a_t^* = 0: \pi^*(0,t) = 1 - \pi_{\min}, \quad \pi^*(\bar{a}_t^*, t) = \pi_{\min}, \quad \pi^*(a,t) = 0 \,\forall a \neq 0, \bar{a}_t^*.$$

## 3 Action-centered contextual bandit

Since the observed reward always contains the sum of the baseline reward and the differential reward we are estimating, and the baseline reward is arbitrarily complex, the main challenge is to isolate the differential reward at each time step. We do this via an action-centering trick, which randomizes the action at each time step, allowing us to construct an estimator whose expectation is proportional to the differential reward $r_t(\bar{a}_t) - r_t(0)$, where $\bar{a}_t$ is the nonzero action chosen by the bandit at time $t$ to be randomized against the zero action. For simplicity of notation, we set the probability of the bandit taking nonzero action $\mathbb{P}(a_t > 0)$ to be equal to $1 - \pi(0,t) = \pi_t$.

### 3.1 Centering the actions - an unbiased $r_t(\bar{a}_t) - r_t(0)$ estimate

To determine a policy, the bandit must learn the coefficients $\theta$ of the model for the differential reward $r_t(\bar{a}_t) - r_t(0) = s_{t,\bar{a}_t}^T \theta$ as a function of $\bar{a}_t$. If the bandit had access at each time $t$ to the differential reward $r_t(\bar{a}_t) - r_t(0)$, we could estimate $\theta$ using a penalized least-squares approach by minimizing

$$\arg \min_\theta \sum_{t=1}^{T} (r_t(\bar{a}_t) - r_t(0) - \theta^T s_{t,\bar{a}_t})^2 + \lambda \|\theta\|_2^2$$

over $\theta$, where $r_t(a)$ is the reward under action $a$ at time $t$ (Agrawal & Goyal, 2013). This corresponds to the Bayesian estimator when the reward is Gaussian. Although we have only access to $r_t(a_t)$, not $r_t(\bar{a}_t) - r_t(0)$, observe that given $\bar{a}_t$, the bandit randomizes to $a_t = \bar{a}_t$ with probability $\pi_t$ and $a_t = 0$ otherwise. Thus

$$\mathbb{E}[(I(a_t > 0) - \pi_t)r_t(a_t)|\mathcal{H}_{t-1}, \bar{a}_t, \bar{s}_t] = \pi_t(1 - \pi_t)r_t(\bar{a}) - (1 - \pi_t)\pi_t r_t(0) \qquad (4)$$
$$= \pi_t(1 - \pi_t)(r_t(\bar{a}_t) - r_t(0)).$$

Thus $(I(a_t > 0) - \pi_t)r_t(a_t)$, which only uses the observed $r_t(a_t)$, is proportional to an unbiased estimator of $r_t(\bar{a}_t) - r_t(0)$. Recalling that $\bar{a}_t, a_t$ are both known since they are chosen by the bandit at time $t$, we create the estimate of the differential reward between $\bar{a}_t$ and action 0 at time $t$ as

$$\hat{r}_t(\bar{a}_t) = (I(a_t > 0) - \pi_t)r_t(a_t).$$

The corresponding penalized weighted least-squares estimator for $\theta$ using $\hat{r}_t(\bar{a}_t)$ is the minimizer of

$$\sum_{t=1}^{T} \pi_t(1 - \pi_t)(\hat{r}_t(\bar{a}_t)/(\pi_t(1 - \pi_t)) - \theta^T s_{t,\bar{a}_t})^2 + \|\theta\|_2^2 \qquad (5)$$
$$= \sum_{t=1}^{T} \frac{(\hat{r}_t(\bar{a}_t))^2}{\pi_t(1 - \pi_t)} - 2\hat{r}_t(\bar{a}_t)\theta^T s_{t,\bar{a}_t} + \pi_t(1 - \pi_t)(\theta^T s_{t,\bar{a}_t})^2 + \|\theta\|_2^2$$
$$= c - 2\theta^T \hat{b} + \theta^T B \theta + \|\theta\|_2^2,$$

where for simplicity of presentation we have used unit penalization $\|\theta\|_2^2$, and

$$\hat{b} = \sum_{t=1}^{T}(I(a_t > 0) - \pi_t)s_{t,\bar{a}_t}r_t(a_t), \quad B = I + \sum_{t=1}^{T}\pi_t(1 - \pi_t)s_{t,\bar{a}_t}s_{t,\bar{a}_t}^T.$$

The weighted least-squares weights are $\pi_t(1 - \pi_t)$, since $\text{var}\left[\frac{\hat{r}_t(\bar{a}_t)}{\pi_t(1-\pi_t)}\Big| \mathcal{H}_{t-1}, \bar{a}_t, \bar{s}_t\right] = \frac{\text{var}[\hat{r}_t(\bar{a}_t)t|\mathcal{H}_{t-1},\bar{a}_t,\bar{s}_t]}{(\pi_t(1-\pi_t))^2}$ and the standard deviation of $\hat{r}_t(\bar{a}_t) = (I(a_t > 0) - \pi_t)r_t(a_t)$ given $\mathcal{H}_{t-1}, \bar{a}_t, \bar{s}_t$ is of order $g_t(\bar{s}_t) = O(1)$. The minimizer of (5) is $\hat{\theta} = B^{-1}\hat{b}$.

## 3.2 Action-Centered Thompson Sampling

As the Thompson sampling approach generates probabilities of taking an action, rather than selecting an action, Thompson sampling is particularly suited to our regression approach. We follow the basic framework of the contextual Thompson sampling approach presented by Agrawal & Goyal (2013), extending and modifying it to incorporate our action-centered estimator and probability constraints.

The critical step in Thompson sampling is randomizing the model coefficients according to the prior $\mathcal{N}(\hat{\theta}, v^2 B^{-1})$ for $\theta$ at time $t$. A $\theta' \sim \mathcal{N}(\hat{\theta}, v^2 B^{-1})$ is generated, and the action $a_t$ chosen to maximize $s_{t,a}^T\theta'$. The probability that this procedure selects any action $a$ is determined by the distribution of $\theta'$; however, it may select action 0 with a probability not in the required range $[1 - \pi_{\max}, 1 - \pi_{\min}]$. We thus introduce a two-step hierarchical procedure. After generating the random $\theta'$, we instead choose the *nonzero* $\bar{a}_t$ maximizing the expected reward

$$\bar{a}_t = \arg\max_{a \in \{1, \ldots, N\}} s_{t,a}^T\theta'.$$

Then we randomly determine whether to take the nonzero action, choosing $a_t = \bar{a}_t$ with probability

---

**Algorithm 1** Action-Centered Thompson Sampling

1: Set $B = I, \hat{\theta} = 0, \hat{b} = 0$, choose $[\pi_{\min}, \pi_{\max}]$.
2: **for** $t = 1, 2, \ldots$ **do**
3:     Observe current context $\bar{s}_t$ and form $s_{t,a}$ for each $a \in \{1, \ldots, N\}$.
4:     Randomly generate $\theta' \sim \mathcal{N}(\hat{\theta}, v^2 B^{-1})$.
5:     Let

$$\bar{a}_t = \arg\max_{a \in \{1, \ldots, N\}} s_{t,a}^T\theta'.$$

6:     Compute probability $\pi_t$ of taking a nonzero action according to (6).
7:     Play action $a_t = \bar{a}_t$ with probability $\pi_t$, else play $a_t = 0$.
8:     Observe reward $r_t(a_t)$ and update $\hat{\theta}$

$$B = B + \pi_t(1 - \pi_t)s_{t,\bar{a}_t}s_{t,\bar{a}_t}^T, \quad \hat{b} = \hat{b} + s_{t,\bar{a}_t}(I(a_t > 0) - \pi_t)r_t(a_t), \quad \hat{\theta} = B^{-1}\hat{b}.$$

9: **end for**

---

$$\pi_t = \mathbb{P}(a_t > 0) = \max(\pi_{\min}, \min(\pi_{\max}, \mathbb{P}(s_{t,\bar{a}}^T\tilde{\theta} > 0))), \tag{6}$$

and $a_t = 0$ otherwise, where $\tilde{\theta} \sim \mathcal{N}(\hat{\theta}, v^2 B^{-1})$. $\mathbb{P}(s_{t,\bar{a}}^T\tilde{\theta} > 0)$ is the probability that the expected relative reward $s_{t,\bar{a}}^T\tilde{\theta}$ of action $\bar{a}_t$ is higher than zero for $\tilde{\theta} \sim \mathcal{N}(\hat{\theta}, v^2 B^{-1})$. This probability is easily computed using the normal CDF. Finally the bandit updates $\hat{b}$, $B$ and computes an updated $\hat{\theta} = B^{-1}\hat{b}$. Our action-centered Thompson sampling algorithm is summarized in Algorithm 1.

## 4 Regret analysis

Classically, the regret of a bandit is defined as the difference between the reward achieved by taking the optimal actions $a_t^*$, and the expected reward received by playing the arm $a_t$ chosen by the bandit

$$\text{regret}_{classical}(t) = s_{t,a_t^*}^T\theta - s_{t,a_t}^T\theta, \tag{7}$$

where the expectation is taken conditionally on $a_t, s_{t,a_t}^T, \mathcal{H}_{t-1}$. For simplicity, let $\pi_t^* = 1 - \pi_t^*(0, t)$ be the probability that the optimal policy takes a nonzero action, and recall that $\pi_t = 1 - \pi_t(0, t)$ is the probability the bandit takes a nonzero action. The probability constraint implies that the optimal policy (3) plays the optimal arm with a probability bounded away from 0 and 1, hence definition (7) is no longer meaningful. We can instead create a regret that is the difference in expected rewards conditioned on $\bar{a}_t, \pi_t, s_{t,a_t}^T, \mathcal{H}_{t-1}$, but not on the randomized action $a_t$:

$$\mathrm{regret}(t) = \pi_t^* s_{t,\bar{a}_t^*}^T \theta - \pi_t s_{t,\bar{a}_t}^T \theta \tag{8}$$

where we have recalled that given $\bar{a}_t$, the bandit plays action $a_t = \bar{a}_t$ with probability $\pi_t$ and plays the $a_t = 0$ with differential reward 0 otherwise. The action-centered contextual bandit attempts to minimize the cumulative regret $\mathcal{R}(T) = \sum_{t=1}^T \mathrm{regret}(t)$ over horizon $T$.

## 4.1 Regret bound for Action-Centered Thompson Sampling

In the following theorem we show that with high probability, the probability-constrained Thompson sampler has low regret relative to the optimal probability-constrained policy.

**Theorem 1.** *Consider the action-centered contextual bandit problem, where $\bar{f}_t$ is potentially time varying, and $\bar{s}_t$ at time $t$ given $\mathcal{H}_{t-1}$ is chosen by an adversary. Under this regime, the total regret at time $T$ for the action-centered Thompson sampling contextual bandit (Algorithm 1) satisfies*

$$\mathcal{R}(T) \leq C \left( \frac{d^2}{\epsilon} \sqrt{T^{1+\epsilon}} (\log(Td) \log \frac{1}{\delta}) \right)$$

*with probability at least $1 - 3\delta/2$, for any $0 < \epsilon < 1$, $0 < \delta < 1$. The constant $C$ is in the proof.*

Observe that this regret bound does not depend on the number of actions $N$, is sublinear in $T$, and scales only with the complexity $d$ of the interaction term, not the complexity of the baseline reward $g$. Furthermore, $\epsilon = 1/\log(T)$ can be chosen giving a regret of order $O(d^2 \sqrt{T})$.

This bound is of the same order as the baseline Thompson sampling contextual bandit in the adversarial setting when the baseline is identically zero (Agrawal & Goyal, 2013). When the baseline can be modeled with $d'$ features where $d' > d$, our method achieves $O(d^2 \sqrt{T})$ regret whereas the standard Thompson sampling approach has $O((d + d')^2 \sqrt{T})$ regret. Furthermore, when the baseline reward is time-varying, the worst case regret of the standard Thompson sampling approach is $O(T)$, while the regret of our method remains $O(d^2 \sqrt{T})$.

## 4.2 Proof of Theorem 1 - Decomposition of the regret

We will first bound the regret (8) at time $t$.

$$\mathrm{regret}(t) = \pi_t^* s_{t,\bar{a}_t^*}^T \theta - \pi_t s_{t,\bar{a}_t}^T \theta = (\pi_t^* - \pi_t)(s_{t,\bar{a}_t}^T \theta) + \pi_t^*(s_{t,\bar{a}_t^*}^T \theta - s_{t,\bar{a}_t}^T \theta) \tag{9}$$

$$\leq (\pi_t^* - \pi_t)(s_{t,\bar{a}_t}^T \theta) + (s_{t,\bar{a}_t^*}^T \theta - s_{t,\bar{a}_t}^T \theta), \tag{10}$$

where the inequality holds since $(s_{t,\bar{a}_t^*}^T \theta - s_{t,\bar{a}_t}^T \theta) \geq 0$ and $0 < \pi_t^* < 1$ by definition. Then

$$\mathcal{R}(T) = \sum_{t=1}^T \mathrm{regret}(t) \leq \underbrace{\sum_{t=1}^T (\pi_t^* - \pi_t)(s_{t,\bar{a}_t}^T \theta)}_{I} + \underbrace{\sum_{t=1}^T (s_{t,\bar{a}_t^*}^T \theta - s_{t,\bar{a}_t}^T \theta)}_{II}$$

Observe that we have decomposed the regret into a term $I$ that depends on the choice of the randomization $\pi_t$ between the zero and nonzero action, and a term $II$ that depends only on the choice of the potential nonzero action $\bar{a}_t$ prior to the randomization. We bound $I$ using concentration inequalities, and bound $II$ using arguments paralleling those for standard Thompson sampling.

**Lemma 1.** *Suppose that the conditions of Theorem 1 apply. Then with probability at least $1 - \frac{\delta}{2}$, $I \leq C \sqrt{d^3 T \log(Td) \log(1/\delta)}$ for some constant $C$ given in the proof.*

**Lemma 2.** *Suppose that the conditions of Theorem 1 apply. Then term $II$ can be bounded as*

$$II = \sum_{t=1}^T (s_{t,\bar{a}_t^*}^T \theta - s_{t,\bar{a}_t}^T \theta) \leq C' \left( \frac{d^2}{\epsilon} \sqrt{T^{1+\epsilon}} \log \frac{1}{\delta} \log(Td) \right)$$

*where the inequality holds with probability at least $1 - \delta$.*

The proofs are contained in Sections 4 and 5 in the supplement respectively. In the derivation, the "pseudo-actions" $\bar{a}_t$ that Algorithm 1 chooses prior to the $\pi_t$ baseline-nonzero randomization correspond to the actions in the standard contextual bandit setting. Note that $I$ involves only $\bar{a}_t$, not $\bar{a}_t^*$, hence it is not surprising that the bound is smaller than that for $II$. Combining Lemmas 1 and 2 via the union bound gives Theorem 1.

## 5 Results

### 5.1 Simulated data

We first conduct experiments with simulated data, using $N = 2$ possible nonzero actions. In each experiment, we choose a true reward generative model $r_t(s, a)$ inspired by data from the HeartSteps study (for details see Section 1.1 in the supplement), and generate two length $T$ sequences of state vectors $s_{t,a} \in \mathbb{R}^{NK}$ and $\bar{s}_t \in \mathbb{R}^L$, where the $\bar{s}_t$ are iid Gaussian and $s_{t,a}$ is formed by stacking columns $I(a = i)[1; \bar{s}_t]$ for $i = 1, \ldots, N$. We consider both nonlinear and nonstationary baselines, while keeping the treatment effect models the same. The bandit under evaluation iterates through the $T$ time points, at each choosing an action and receiving a reward generated according to the chosen model. We set $\pi_{\min} = 0.2, \pi_{\max} = 0.8$.

At each time step, the reward under the optimal policy is calculated and compared to the reward received by the bandit to form the regret $\text{regret}(t)$. We can then plot the cumulative regret

$$\text{cumulative regret}(t) = \sum_{\tau=1}^{t} \text{regret}(\tau).$$

In the first experiment, the baseline reward is nonlinear. Specifically, we generate rewards using

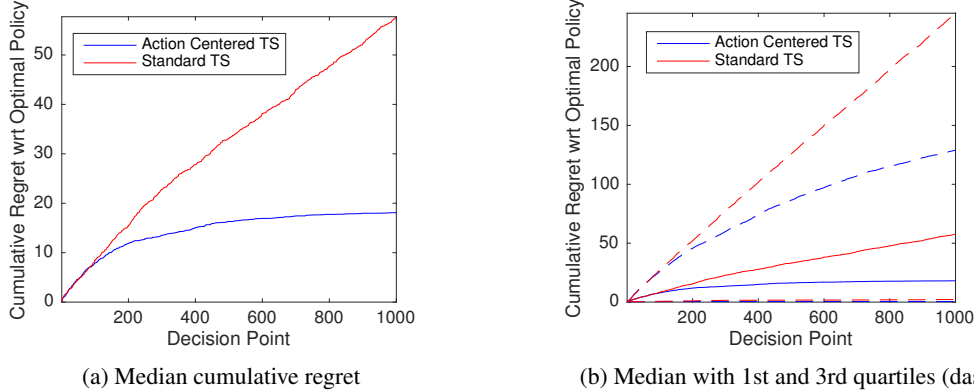

|     |     |
| --- | --- |
| (a) Median cumulative regret | (b) Median with 1st and 3rd quartiles (dashed) |

Figure 1: Nonlinear baseline reward $g$, in scenario with 2 nonzero actions and reward function based on collected HeartSteps data. Cumulative regret shown for proposed Action-Centered approach, compared to baseline contextual bandit, median computed over 100 random trials.

$r_t(s_{t,a_t}, \bar{s}_t, a_t) = \theta^T s_{t,a_t} + 2I(|[\bar{s}_t]_1| < 0.8) + n_t$ where $n_t = \mathcal{N}(0, 1)$ and $\theta \in \mathbb{R}^8$ is a fixed vector listed in supplement section 1.1. This simulates the quite likely scenario that for a given individual the baseline reward is higher for small absolute deviations from the mean of the first context feature, i.e. rewards are higher when the feature at the decision point is "near average", with reward decreasing for abnormally high or low values. We run the benchmark Thompson sampling algorithm (Agrawal & Goyal, 2013) and our proposed action-centered Thompson sampling algorithm, computing the cumulative regrets and taking the median over 500 random trials. The results are shown in Figure 1, demonstrating linear growth of the benchmark Thompson sampling algorithm and significantly lower, sublinear regret for our proposed method.

We then consider a scenario with the baseline reward $g_t(\cdot)$ function changing in time. We generate rewards as $r_t(s_{t,a_t}, \bar{s}_t, a_t) = \theta^T s_{t,a_t} + \eta_t^T \bar{s}_t + n_t$ where $n_t = \mathcal{N}(0, 1)$, $\theta$ is a fixed vector as above, and $\eta_t \in \mathbb{R}^7$, $\bar{s}_t$ are generated as smoothly varying Gaussian processes (supplement Section 1.1). The cumulative regret is shown in Figure 2, again demonstrating linear regret for the baseline approach and significantly lower sublinear regret for our proposed action-centering algorithm as expected.

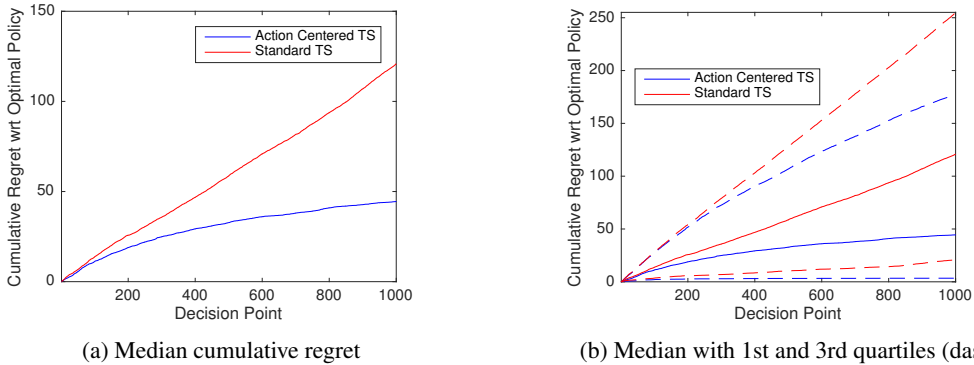

| (a) Median cumulative regret | (b) Median with 1st and 3rd quartiles (dashed) |

Figure 2: Nonstationary baseline reward $g$, in scenario with 2 nonzero actions and reward function based on collected HeartSteps data. Cumulative regret shown for proposed Action-Centered approach, compared to baseline contextual bandit, median computed over 100 random trials.

## 5.2 HeartSteps study data

The HeartSteps study collected the sensor and weather-based features shown in Figure 1 at 5 decision points per day for each study participant. If the participant was available at a decision point, a message was sent with constant probability 0.6. The sent message could be one of several activity or anti-sedentary messages chosen by the system. The reward for that message was defined to be $\log(0.5 + x)$ where $x$ is the step count of the participant in the 30 minutes following the suggestion. As noted in the introduction, the baseline reward, i.e. the step count of a subject when no message is sent, does not only depend on the state in a complex way but is likely dependent on a large number of unobserved variables. Because of these unobserved variables, the mapping from the observed state to the reward is believed to be strongly time-varying. Both these characteristics (complex, time-varying baseline reward function) suggest the use of the action-centering approach.

We run our contextual bandit on the HeartSteps data, considering the binary action of whether or not to send a message at a given decision point based on the features listed in Figure 1 in the supplement. Each user is considered independently, for maximum personalization and independence of results. As above we set $\pi_{\min} = 0.2, \pi_{\max} = 0.8$.

We perform offline evaluation of the bandit using the method of Li et al. (2011). Li et al. (2011) uses the sequence of states, actions, and rewards in the data to form an near-unbiased estimate of the average expected reward achieved by each algorithm, averaging over all users. We used a total of 33797 time points to create the reward estimates. The resulting estimates for the improvement in average reward over the baseline randomization, averaged over 100 random seeds of the bandit algorithm, are shown in Figure 2 of the supplement with the proposed action-centering approach achieving the highest reward. Since the reward is logarithmic in the number of steps, the results imply that the benchmark Thompson sampling approach achieves an average 1.6% increase in step counts over the non-adaptive baseline, while our proposed method achieves an increase of 3.9%.

## 6 Conclusion

Motivated by emerging challenges in adaptive decision making in mobile health, in this paper we proposed the action-centered Thompson sampling contextual bandit, exploiting the randomness of the Thompson sampler and an action-centering approach to orthogonalize out the baseline reward. We proved that our approach enjoys low regret bounds that scale only with the complexity of the interaction term, allowing the baseline reward to be arbitrarily complex and time-varying.

## Acknowledgments

This work was supported in part by grants R01 AA023187, P50 DA039838, U54EB020404, R01 HL125440 NHLBI/NIA, NSF CAREER IIS-1452099, and a Sloan Research Fellowship.

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
