[Supplementary Material]

# Supplement to Action Centered Contextual Bandits

**Kristjan Greenewald**
Department of Statistics
Harvard University
kgreenewald@fas.harvard.edu

**Ambuj Tewari**
Department of Statistics
University of Michigan
tewaria@umich.edu

**Predrag Klasnja**
School of Information
University of Michigan
klasnja@umich.edu

**Susan Murphy**
Departments of Statistics and Computer Science
Harvard University
samurphy@fas.harvard.edu

## 1 HeartSteps feature list

Figure 1 shows the features available to the bandit in the HeartSteps study dataset, and Figure 2 shows the estimated average regret results with errorbars.

| Feature | Description | Purpose | Interaction | Baseline Model |
|---|---|---|---|---|
| Number of messages sent | Total number of messages sent to user in prior week | Modeling habituation to intervention | Y | Y |
| Location indicator 1 | 1 if not at home or work, 0 o.w. | Location relevant to availability to walk | Y | Y |
| Location indicator 2 | 1 if at work, 0 o.w. | | Y | Y |
| Step count variability | Historical standard deviation of step counts in 60 minute window surrounding decision point, taken over prior 7 days | Responsiveness in different times of day | Y | Y |
| Steps in prior 30 minutes | Step count in 30 minutes prior to decision point | Measure of recent activity | | Y |
| Square root of steps yesterday | Square root of the total step count yesterday | Recent commitment/ engagement | | Y |
| Outdoor Temperature | Degrees Celsius | Cold weather potentially less appealing | | Y |

Figure 1: List of features available to the bandit in the HeartSteps experiment. The features available to model the action interaction (effect of sending an anti-sedentary message) and to model the baseline (reward under no action) are denoted via a "Y" in the corresponding column.

### 1.1 Simulation model

Figure 3 shows the coefficients $\theta$ used in the main text simulations. The coefficients shown in the figure associated with the first action are obtained via a linear regression analysis of the binary action (sending or not sending a message) HeartSteps intervention data, and the coefficients for the second action are a simple modification of those.

For the time varying simulation, Gaussian processes were used to generate the reward coefficient sequence $\eta_t$ and the state sequence $\bar{s}_t$. We used Gaussian processes since if $\eta_t$ is IID, then the baseline reward becomes an IID random variable, making the baseline reward not time varying.

We used the Gaussian process

$$\eta_t = \sqrt{1-\rho^2}\eta_{t-1} + \rho n_t$$

where $\eta_0 = 1_7$, $n_t \sim \mathcal{N}(0, I_7)$, and $\rho = 0.1$. The state sequence $\bar{s}_t$ was generated in the same manner.

Figure 2: Unbiased estimates of the average reward received by the benchmark Thompson sampling contextual bandit and the proposed action-centered Thompson sampling contextual bandit, relative to the reward received under the pre-specified HeartSteps randomization policy. Also shown are one standard deviation error bars for the computed estimates. The superior performance of the action-centering approach is indicative of its robustness to the high complexity of the baseline subject behavior.

| Feature | Action 1 coef. | Action 2 coef. |
|---|---|---|
| Number of messages sent | .116 | .116 |
| Location indicator 1 | -.275 | .275 |
| Location indicator 2 | -.233 | -.233 |
| Step count variability | .0425 | .0425 |

Figure 3: Effect coefficients, based on HeartSteps data, used for simulation reward model.

## 2  Definitions

In order to proceed with the proof of Theorem 1, we make the following definitions.

**Definition 1.** *Define a filtration $\mathcal{F}_{t-1} = \{\mathcal{H}_{t-1}, \bar{s}_t\}$ as the union of the history and current context.*

**Definition 2.** *Let*
$$z_{t,a} = \sqrt{s_{t,a}^T B(t)^{-1} s_{t,a}},$$
*for all $a = 1, \ldots, N$.*

**Definition 3.** *Define $\ell(T) = R\sqrt{d\log(T^3)\log(1/\delta)} + 1$, $v = R\sqrt{\frac{24}{\epsilon} d\log(1/\delta)}$, and $g(T) = \sqrt{4d\log(Td)} v + \ell(T)$.*

We divide the arms $\bar{a} > 0$ into saturated and unsaturated actions.

**Definition 4** (Saturated vs. unsaturated actions). *Any arm $\bar{a} > 0$ for which $g(T)z_{t,\bar{a}} < \ell(T)z_{t,\bar{a}_t^*}$ is called a* saturated *arm. If an arm is not saturated, it is called* unsaturated. *Let $C(t) \subseteq \{1, \ldots, N\}$ be the subset of saturated arms at time $t$.*

Observe that the optimal arm $\bar{a}^*$ is unsaturated by definition.

We can now state the required concentration events and present bounds on the probability they occur.

### 2.1  Concentration events

**Definition 5.** *Let $E^\mu(t)$ be the event that for all $\bar{a} = 1, \ldots, N$*
$$|s_{t,\bar{a}}^T \hat{\theta}_t - s_{t,\bar{a}}^T \theta| \leq \ell(T)z_{t,\bar{a}}.$$

*Similarly, let $E^\theta(t)$ be the event that for all $\bar{a} = 1, \ldots, N$*

$$|s_{t,\bar{a}}^T \theta'_t - s_{t,\bar{a}}^T \hat{\theta}_t| \leq \sqrt{4d \log(Td)} v z_{t,\bar{a}}$$

*and $E_0^\theta(t)$ be the corresponding event that for all $\bar{a} = 1, \ldots, N$*

$$|s_{t,\bar{a}}^T \tilde{\theta}_t - s_{t,\bar{a}}^T \hat{\theta}_t| \leq \sqrt{4d \log(Td)} v z_{t,\bar{a}}$$

We can bound the probabilities of the events $E^\theta(t)$, $E^\theta(t)_0$, and $E^\mu(t)$ in the following lemmas. Observe that by definition $\mathbb{P}(E^\theta(t)|\mathcal{F}_{t-1}) = \mathbb{P}(E^\theta(t)_0|\mathcal{F}_{t-1})$.

**Lemma 1** (Agrawal & Goyal (2013)). *For all t, and possible filtrations $\mathcal{F}_{t-1}$, $\mathbb{P}(E^\theta(t)|\mathcal{F}_{t-1}) \geq 1 - \frac{1}{T^2}$.*

For $E^\mu(t)$ we have

**Lemma 2.** *For all t, $0 < \delta < 1$, $\mathbb{P}(E^\mu(t)) \geq 1 - \frac{\delta}{T^2}$.*

The proof is given in Section 7.

## 2.2 Supermartingales

**Definition 6** (Supermartingale). *A sequence of random variables $(Y_t; t \geq 0)$ is called a supermartingale corresponding to a filtration $\mathcal{F}_t$ if, for all t, $Y_t$ is $\mathcal{F}_t$-measurable, and*

$$\mathbb{E}[Y_t - Y_{t-1}|\mathcal{F}_{t-1}] \leq 0$$

*for all $t \geq 1$.*

**Lemma 3** (Azuma-Hoeffding inequality). *If for all $t = 1, \ldots, T$ a supermartingale $(Y_t; t \geq 0)$ corresponding to filtration $\mathcal{F}_t$ satisfies $|Y_t - Y_{t-1}| \leq c_t$ for some constants $c_t$, then for any $a \geq 0$*

$$\mathbb{P}(Y_T - Y_0 \geq 0) \leq e^{-\frac{a^2}{2\sum_{t=1}^T c_t^2}}.$$

# 3 Preliminary results

## 3.1 Lemma 5: Probability of choosing a saturated action $\bar{a}_t \in C(t)$

**Lemma 4** (Agrawal & Goyal (2013) Lemma 2). *For any filtration $\mathcal{F}_{t-1}$ such that $E^\mu(t)$ is true,*

$$\mathbb{P}(s_{t,\bar{a}_t^*}^T \theta' > s_{t,\bar{a}_t^*}^T \theta + \ell(T) z_{t,\bar{a}_t^*}|\mathcal{F}_{t-1}) \geq \frac{1}{4e\sqrt{\pi T^\epsilon}}.$$

We can now prove the following.

**Lemma 5.** *For any filtration $\mathcal{F}_{t-1}$ such that $E^\mu(t)$ is true,*

$$\mathbb{P}(\bar{a}_t \in C(t)|\mathcal{F}_{t-1}) \leq \frac{1}{p}\mathbb{P}(\bar{a}_t \notin C(t)|\mathcal{F}_{t-1}) + \frac{1}{pT^2},$$

*where $p = \frac{1}{4e\sqrt{\pi T^\epsilon}}$.*

*Proof.* Recall that $\bar{a}_t$ is the action with the largest value of $s_{t,i}^T \theta'$. Hence, if $s_{t,\bar{a}_t^*}^T \theta'$ is larger than $s_{t,i}^T \theta'$ for all $i \in C(t)$, then $\bar{a}_t$ is one of the unsaturated actions. Hence

$$\mathbb{P}(\bar{a}_t \notin C(t)|\mathcal{F}_{t-1}) \geq \mathbb{P}(s_{t,\bar{a}_t^*}^T \theta' > s_{t,i}^T \theta', \forall i \in C(t)|\mathcal{F}_{t-1}. \tag{1}$$

We know that by definition all saturated arms $i \in C(t)$ have $g(T)z_{t,j} < \ell(T)z_{t,\bar{a}_t^*}$. Given an $\mathcal{F}_{t-1}$ such that $E^\mu(t)$ holds, we have that either $E^\theta(t)$ is false or for all $i \in C(t)$

$$s_{t,i}^T \theta' \leq s_{t,i}^T \theta + g(T)z_{t,i} \leq s_{t,\bar{a}_t^*}^T \theta + \ell(T)z_{t,\bar{a}_t^*}$$

implying

$$\mathbb{P}(s_{t,\bar{a}_t^*}^T \theta' > s_{t,i}^T \theta', \forall j \in C(t)|\mathcal{F}_{t-1})$$
$$\geq \mathbb{P}(s_{t,\bar{a}_t^*}^T \theta' > s_{t,\bar{a}_t^*}^T \theta + \ell(T)z_{t,\bar{a}_t^*}|\mathcal{F}_{t-1}) - \mathbb{P}(\overline{E^\theta(t)}|\mathcal{F}_{t-1})$$
$$\geq p - \frac{1}{T^2}.$$

where we have used the definitions of $E^\mu(t)$, $E^\theta(t)$, and the last inequality follows from Lemma 4 and Lemma 2. Substituting into (1) gives

$$\mathbb{P}(\bar{a}_t \notin C(t)|\mathcal{F}_{t-1}) + \frac{1}{T^2} \geq p,$$

and

$$\frac{\mathbb{P}(\bar{a}_t \in C(t)|\mathcal{F}_{t-1})}{\mathbb{P}(\bar{a}_t \notin C(t)|\mathcal{F}_{t-1}) + \frac{1}{T^2}} \leq \frac{1}{p}.$$

$\square$

## 3.2 Lemma 7 - Bound on $\sum_t z_{t,\bar{a}_t}$

**Lemma 6.** *For* $z_{t,a} = \sqrt{s_{t,a}^T B(t)^{-1} s_{t,a}}$, *we have that*

$$\sum_{t=1}^T z_{t,\bar{a}_t} \leq \frac{5}{C_\pi}\sqrt{dT \log T},$$

*where* $C_\pi = \sqrt{\min(\pi_{\min}(1-\pi_{\max}), \pi_{\max}(1-\pi_{\min})}$ *is a contant.*

*Proof.* We apply the following lemma from Auer et al. (2002) and Chu et al. (2011).

**Lemma 7.** *Let* $A_t = I + \sum_{t=1}^T x_t x_t^T$, *where* $x_t \in \mathbb{R}^d$ *is a sequence of vectors. Then, defining* $\sigma_t = \sqrt{x_t^T A_t^{-1} x_t}$, *we have*

$$\sum_{t=1}^T \sigma_t \leq 5\sqrt{dT \log T}.$$

To apply this to $\sum_t z_{t,\bar{a}_t}$, let $x_t = \sqrt{\pi_t(1-\pi_t)}s_{t,\bar{a}_t}$. Then $A_t = I + \sum_{t=1}^T (\pi_t(1-\pi_t))s_{t,\bar{a}_t}s_{t,\bar{a}_t}^T = B_t$, and we have

$$\sigma_t = \sqrt{x_t^T A_t^{-1} x_t} = \sqrt{\pi_t(1-\pi_t)}\sqrt{s_{t,\bar{a}_t}^T B_t s_{t,\bar{a}_t}} = \sqrt{\pi_t(1-\pi_t)}z_{t,\bar{a}_t}.$$

Applying Lemma 7 we thus have

$$\sum_{t=1}^T z_{t,\bar{a}_t} \leq \max_t \left(\frac{1}{\sqrt{\pi_t(1-\pi_t)}}\right) \sum_{t=1}^T \sigma_t \leq \frac{5}{C_\pi}\sqrt{dT \log T},$$

where $C_\pi = \sqrt{\min(\pi_{\min}(1-\pi_{\max}), \pi_{\max}(1-\pi_{\min})}$ is a constant. $\square$

# 4 Proof of Lemma 1 - term I

*Proof.* We know that by definition of the optimal policy, $(\pi_t^* - \pi_t)s_{t,\bar{a}_t}^T \theta \geq 0$. Hence under event $E^\mu(t)$,

$$(\pi_t^* - \pi_t)s_{t,\bar{a}_t}^T \theta \leq \mathbb{P}\left(\text{sign}(s_{t,\bar{a}_t}^T \theta') \neq \text{sign}(s_{t,\bar{a}_t}^T \theta))\right) |s_{t,\bar{a}_t}^T \theta|$$
$$\leq \min\left[|s_{t,\bar{a}_t}^T \theta|, \mathbb{P}(\text{sign}(s_{t,\bar{a}_t}^T \theta') \neq \text{sign}(s_{t,\bar{a}_t}^T \theta))\right]$$
$$\leq (\ell(T) + \sqrt{4d \log(Td)}v)z_{t,\bar{a}_t} + 1 - \mathbb{P}(E_0^\theta(t)).$$

Substituting in the definitions of $\ell(T), v$ and the bound in Lemma 1 on $\mathbb{P}(E_0^\theta(t))$, we have

$$(\pi_t^* - \pi_t)s_{t,\bar{a}_t}^T \theta \leq \left( R\sqrt{d \log(T^3) \log(1/\delta)} + 1 + \sqrt{4d \log(Td)} R\sqrt{\frac{24}{\epsilon} d \log(1/\delta)} \right) z_{t,\bar{a}_t} + \frac{1}{T^2}$$

$$\leq C\sqrt{\frac{d^2}{\epsilon} \log(1/\delta)} z_{t,\bar{a}_t} + \frac{1}{T^2}.$$

Summing over $t$ and recalling that by Lemma 7 $\sum_{t=1}^T z_{t,\bar{a}_t} \leq \frac{5}{C_\pi} \sqrt{dT \log T}$, we have that under event $E^\mu(t)$

$$I = \sum_{t=1}^T (\pi_t^* - \pi_t)s_{t,\bar{a}_t}^T \theta$$

$$\leq \frac{C}{C_\pi} \sqrt{d^3 T \log(Td) \log(1/\delta)}.$$

Since the probability that $E^\mu(t)$ holds is at least $1 - \frac{\delta}{T^2}$ by Lemma 2, the lemma results. $\qquad \square$

## 5 Proof of Lemma 2: Bound on term $II$

Before commencing the proof, we first state the following result from Abbasi-Yadkori et al. (2011).

**Lemma 8** (Abbasi-Yadkori et al. (2011)). *Let $(\mathcal{F}_t'; t \geq 0)$ be a filtration, $(m_t; t \geq 1)$ be an $\mathbb{R}^d$-valued stochastic process such that $m_t$ is $(\mathcal{F}_{t-1}')$- measurable, $(\eta_t; t \geq 1)$ be a real-valued martingale difference process such that $\eta_t$ is $(\mathcal{F}_t')$-measurable. For $t \geq 0$, define $\xi_t = \sum_{\tau=1}^t m_\tau \eta_\tau$ and $M_t = I_d + \sum_{\tau=1}^t m_\tau m_\tau^T$, where $I_d$ is the d-dimensional identity matrix. Assume $\eta_t$ is conditionally R-sub-Gaussian.*

*Then, for any $\delta' > 0$, $t \geq 0$, with probability at least $1 - \delta'$,*

$$\|\xi_t\|_{M_t^{-1}} \leq R\sqrt{d \log\left(\frac{t+1}{\delta'}\right)},$$

*where $\|\xi_t\|_{M_t^{-1}} = \sqrt{\xi_t^T M_t^{-1} \xi_t}$.*

We now prove Lemma 2.

*Proof.* Defining $\text{regret}'(t) = (s_{t,\bar{a}_t^*}^T \theta - s_{t,\bar{a}_t}^T \theta)I(E^\mu(t))$, we have the following lemma, which we prove in Section 6.

**Lemma 9.** *Let, for $p = \frac{1}{4e\sqrt{\pi T^\epsilon}}$,*

$$X_t = \text{regret}'(t) - \frac{g(T)}{p} I(a(t) \notin C(t)) z_{t,\bar{a}_t^*} \tag{2}$$

$$Y_t = \sum_{w=1}^t X_w. \tag{3}$$

*Then $(Y_t; t = 0, \ldots, T)$ is a super-martingale process with respect to filtration $\mathcal{F}_t$.*

Given our results in Section 7.1 and our concentration bounds, the proof is closely related to Agrawal & Goyal (2013) and is listed in Section 6.

Using the definition of $X_t$, we have that $|Y_t - Y_{t-1}| \leq |X_t| \leq 1 + \frac{g(T)}{p} + \frac{2g(T)^2}{\ell(T)} + \frac{2g(T)}{pT^2} \leq \frac{8}{p}\frac{g(T)^2}{\ell(T)}$. This allows us to apply the Azuma-Hoeffding inequality listed in Section 2.2, giving that

$$
\sum_{t=1}^{T} \text{regret}'(t) \leq \sum_{t=1}^{T}\left(\frac{g(T)}{p}I(\bar{a}_t \notin C(t))z_{t,\bar{a}_t^*}\right) + \frac{2g(T)}{pT} + \frac{2g(T)^2}{\ell(T)}\sum_{t=1}^{T}z_{t,\bar{a}_t} + \frac{8}{p}\frac{g(T)^2}{\ell(T)}\sqrt{2T\log\frac{2}{\delta}}
$$

$$
\leq \sum_{t=1}^{T}\left(\frac{g(T)^2}{\ell(T)}\frac{1}{p}I(\bar{a}_t \notin C(t))z_{t,\bar{a}_t}\right) + \frac{2g(T)}{pT} + \frac{2g(T)^2}{\ell(T)}\sum_{t=1}^{T}z_{t,\bar{a}_t} + \frac{8}{p}\frac{g(T)^2}{\ell(T)}\sqrt{2T\log\frac{2}{\delta}}
$$

$$
\leq \frac{g(T)^2}{\ell(T)}\frac{3}{p}\sum_{t=1}^{T}z_{t,\bar{a}_t} + \frac{2g(T)}{pT} + \frac{8}{p}\frac{g(T)^2}{\ell(T)}\sqrt{2T\log\frac{2}{\delta}}.
$$

with probability at least $1 - \delta/2$, where we recall that if $\bar{a}_t \notin C(t)$, then $g(T)z_{t,\bar{a}_t} \geq \ell(T)z_{t,\bar{a}_t^*}$.

Substituting in the bound $\sum_{t=1}^{T} z_{t,\bar{a}_t} \leq \frac{5}{C_\pi}\sqrt{dT\log T}$ from Lemma 7 and the definitions of $g(T), p, \ell(T)$, we obtain that

$$
\sum_{t=1}^{T}\text{regret}'(t) \leq \frac{C'}{C_\pi}\left(\frac{d^2}{\epsilon}\sqrt{T^{1+\epsilon}}\log\frac{1}{\delta}\log(Td)\right)
$$

with probability at least $1 - \frac{\delta}{2}$, where $C'$ is a constant. Recall that by Lemma 2, $E^\mu(t)$ holds for all $t$ with probability at least $1 - \delta/2$, and that $\text{regret}'(t) = (s_{t,\bar{a}_t^*}^T\theta - s_{t,\bar{a}_t}^T\theta)$ whenever $E^\mu(t)$ holds. By the union bound we then have that

$$
II = \sum_{t=1}^{T}(s_{t,\bar{a}_t^*}^T\theta - s_{t,\bar{a}_t}^T\theta) \leq \frac{C'}{C_\pi}\left(\frac{d^2}{\epsilon}\sqrt{T^{1+\epsilon}}\log\frac{1}{\delta}\log(Td)\right)
$$

with probability at least $1 - \delta$. The lemma results.

$\square$

# 6  Proof of Lemma 9

*Proof.* To prove that $Y_t$ is a super-martingale by the definition above, we need to prove that for all $1 \leq t \leq T$ and any $\mathcal{F}_{t-1}$, $\mathbb{E}[Y_t - Y_{t-1}|\mathcal{F}_{t-1}] \leq 0$.

We first consider filtrations $\mathcal{F}_{t-1}$ for which $E^\mu(t)$ holds. By the definition of $\bar{a}_t$, $s_{t,\bar{a}_t}^T\theta' \geq s_{t,a_t^{**}}^T\theta'$. Under $E^\theta(t)$ and $E^\mu(t)$ we then must have that for all $i = 1, \ldots, N$

$$
s_{t,i}^T\theta \geq s_{t,i}^T\theta' - g(T)z_{t,i}
$$
$$
\geq s_{t,a_t^{**}}^T\theta' - g(T)z_{t,i}
$$
$$
\geq s_{t,a_t^{**}}^T\theta - g(T)z_{t,\bar{a}_t^*} - g(T)z_{t,i}.
$$

Hence $s_{t,\bar{a}_t^*}^T\theta - s_{t,\bar{a}_t}^T\theta \leq g(T)(z_{t,\bar{a}_t} + z_{t,\bar{a}_t^*})$.

For $\mathcal{F}_{t-1}$ such that $E^\mu(t)$ holds, we then can write

$$
\mathbb{E}[\text{regret}'(t)|\mathcal{F}_{t-1}] = \mathbb{E}[(s_{t,\bar{a}_t^*}^T\theta - s_{t,\bar{a}_t}^T\theta)|\mathcal{F}_{t-1}]
$$
$$
\geq \mathbb{E}[g(T)(z_{t,\bar{a}_t} + z_{t,\bar{a}_t^*})|\mathcal{F}_{t-1}] + \mathbb{P}(\overline{E^\theta(t)})
$$
$$
= g(T)z_{t,\bar{a}_t^*}\mathbb{P}(\bar{a}_t \in C(t)|\mathcal{F}_{t-1}) + g(T)\mathbb{E}\left[(\frac{g(T)}{\ell(T)}z_{t,\bar{a}_t}I(\bar{a}_t \notin C(t))|\mathcal{F}_{t-1}\right]
$$
$$
+ g(T)\mathbb{E}[z_{t,\bar{a}_t}|\mathcal{F}_{t-1}] + \frac{1}{T^2}.
$$

where we have used the facts that $\text{regret}'(t) \leq 1$, the definition of unsaturated arms, and Lemma 2. Applying Lemma 5 and noting that since $\min \text{eig}(B(t)) \leq 1$, $z_{t,i} \leq \|s_{t,i}\|_2 \leq 1$, we can show that

$$
\mathbb{E}[\text{regret}'(t)|\mathcal{F}_{t-1}] \leq \frac{g(T)}{p}\mathbb{P}(\bar{a}_t \notin C(t)|\mathcal{F}_{t-1})z_{t,\bar{a}_t^*} + \frac{2g(T)}{pT^2}. \tag{4}
$$

By definition, $\text{regret}'(t) = (s_{t,\bar{a}_t^*}^T \theta - s_{t,\bar{a}_t}^T \theta) I(E^\mu(t))$ is zero and the above inequality holds whenever $E^\mu(t)$ is not true. Since we have considered both cases, the lemma is proved.

$\square$

# 7   Proof of Lemma 2

*Proof.* We can apply Lemma 8 with $m_t = \sqrt{\pi_t(1 - \pi_t)} s_{t,\bar{a}_t}$,

$$\eta_t = \frac{\hat{r}_t(\bar{a}_t)}{\sqrt{\pi_t(1 - \pi_t)}} - \sqrt{\pi_t(1 - \pi_t)} s_{t,\bar{a}_t}^T \theta,$$

and with the filtration $\mathcal{F}_t' = (\bar{s}_{\tau+1}, m_{\tau+1}, \eta_\tau : \tau \leq t)$ effectively containing all the available information up to the current time. $\mathcal{F}_{t-1}'$ is measurable by definition, and in Section 7.1 we show

**Lemma 10.** *Suppose that $n_t$ is $R$ sub-Gaussian. Then $\eta_t$ is a $\mathcal{F}_t'$-measurable, $R'$-sub-Gaussian, martingale difference process where $R' = \frac{R+2}{\sqrt{\pi_{\min}(1-\pi_{\max})}} + \sqrt{\pi_{\max}(1 - \pi_{\min})}$.*

We then have

$$M_t = I_d + \sum_{\tau=1}^{t} m_\tau m_\tau^T = I_d + \sum_{\tau=1}^{t} \pi_\tau(1 - \pi_\tau) s_{\tau,\bar{a}_\tau} s_{\tau,\bar{a}_\tau}^T,$$

$$\xi_t = \sum_{\tau=1}^{t} m_\tau \eta_\tau = \sum_{\tau=1}^{t} s_{\tau,\bar{a}_\tau} \left( \hat{r}_t(\bar{a}_t) - \pi_t(1 - \pi_t) s_{t,\bar{a}_t}^T \theta \right).$$

Observe that these are the two primary components of the contextual bandit, specifically, $B_t = M_{t-1}$ and $b_t - \mathbb{E}[b_t] = \xi_t$. Hence, $\hat{\theta}_t - \theta = M_{t-1}^{-1}(\xi_{t-1} - \theta)$. Letting $\|y\|_A = \sqrt{y^T A y}$ for any vector $y$ and matrix $A \in \mathbb{R}^{d \times d}$, for all $\bar{a} > 0$ we have that since $M_t$ is positive definite,

$$|s_{\bar{a},t}^T \hat{\theta} - s_{\bar{a},t}^T \theta| = |s_{\bar{a},t}^T M_{t-1}^{-1}(\xi_{t-1} - \theta)|$$
$$\leq \|s_{\bar{a},t}\|_{M_{t-1}^{-1}} \|\xi_{t-1} - \theta\|_{M_{t-1}^{-1}}$$
$$= \|s_{\bar{a},t}\|_{B_t^{-1}} \|\xi_{t-1} - \theta\|_{B_t^{-1}}.$$

Applying Lemma 8, we have that for any $\delta' > 0$, $t \geq 1$,

$$\|\xi_{t-1}\|_{M_{t-1}^{-1}} \leq R' \sqrt{d \log \frac{t}{\delta'}}.$$

Then $\|\xi_{t-1} - \theta\|_{M_{t-1}^{-1}} \leq R' \sqrt{d \log \frac{t}{\delta'}} + \|\theta\|_{M_{t-1}^{-1}} \leq R' \sqrt{d \log \frac{T}{\delta'}} + 1$. Setting $\delta' = \delta/T^2$ implies that with probability $1 - \delta/T^2$, for all $\bar{a}$,

$$|s_{\bar{a},t}^T \hat{\theta} - s_{\bar{a},t}^T \theta| \leq \|s_{\bar{a},t}\|_{B_t^{-1}} \left( R' \sqrt{d \log(T^3) \log \frac{1}{\delta}} + 1 \right) = \ell(T) z_{t,\bar{a}}.$$

$\square$

## 7.1 Proof of Lemma 10: Martingale analysis of $\eta_t$

*Proof.* Recall

$$
\begin{aligned}
|\eta_t| &= \left| \frac{\hat{r}_t(\bar{a}_t)}{\sqrt{\pi_t(1-\pi_t)}} - \sqrt{\pi_t(1-\pi_t)}s_{t,\bar{a}_t}^T\theta \right| \\
&= \left| \frac{(I(a_t>0)-\pi_t)r_t(a_t)}{\sqrt{\pi_t(1-\pi_t)}} - \sqrt{\pi_t(1-\pi_t)}s_{t,\bar{a}_t}^T\theta \right| \\
&= \left| \frac{(I(a_t>0)-\pi_t)(s_{t,a_t}^T\theta I(a_t>0) + n_t + \bar{f}_t(\bar{s}_t))}{\sqrt{\pi_t(1-\pi_t)}} - \sqrt{\pi_t(1-\pi_t)}s_{t,\bar{a}_t}^T\theta \right| \\
&\leq \sqrt{\pi_t(1-\pi_t)} + \left| \frac{2+n_t}{\sqrt{\pi_t(1-\pi_t)}} \right|.
\end{aligned}
$$

since the rewards are all bounded by one and the $\pi_{\min} \leq \pi_t \leq \pi_{\max}$ are bounded. We have assumed that $n_t$ is R sub-Gaussian. Since a bounded random variable $|X| < b$ is $b$ sub-Gaussian and the sum of independent $b_1$ and $b_2$ sub-Gaussian random variables is $b_1 + b_2$ sub-Gaussian, we have that $\eta_t$ is $R' = \frac{R+2}{\sqrt{\pi_{\max}(1-\pi_{\min})}} + \sqrt{\pi_{\min}(1-\pi_{\max})}$ conditionally sub-Gaussian. Since $\pi_{\min}, \pi_{\max}$ are bounded away from 0 and 1 by constants, $R'$ is a constant.

Additionally, for all $\bar{a}_t$

$$
\begin{aligned}
\mathbb{E}[\eta_t|\mathcal{H}_{t-1},\bar{a}_t,\bar{s}_t] &= \frac{\mathbb{E}[\hat{r}_t(\bar{a}_t)|\mathcal{H}_{t-1},\bar{a}_t,\bar{s}_t]}{\sqrt{\pi_t(1-\pi_t)}} - \sqrt{\pi_t(1-\pi_t)}s_{t,\bar{a}_t}^T\theta \\
&= \frac{\mathbb{E}[(I(a_t>0)-\pi_t)r_t(a_t)|\mathcal{H}_{t-1},\bar{a}_t,\bar{s}_t]}{\sqrt{\pi_t(1-\pi_t)}} - \sqrt{\pi_t(1-\pi_t)}s_{t,\bar{a}_t}^T\theta \\
&= \frac{\pi_t(1-\pi_t)s_{t,\bar{a}_t}^T\theta}{\sqrt{\pi_t(1-\pi_t)}} - \sqrt{\pi_t(1-\pi_t)}s_{t,\bar{a}_t}^T\theta \\
&= 0,
\end{aligned}
$$

where the third equality follows from (4). Thus $\mathbb{E}[\eta_t|\mathcal{H}_{t-1},\bar{s}_t] = 0$ and $\eta_t$ is a martingale difference process.

$\square$