[Reviews · NeurIPS 2017]

Reviewer 1



This paper considers the problem of contextual bandits where the dependency for one action maybe arbitrarily complex, but the relative expected rewards of other actions might be simple. This is an attractive framework for problems with "default" actions, such those considered by the authors. Hence, I recommend acceptance even though the model is very simple and is a nearly direct application of existing results. However, the authors should clarify the links with the restless bandit literature (e.g. [1]) and the non-linear bandit literature (e.g. [2]). The hypothesis to be tested is whether or not the algorithm is better when the modelling assumptions are satisfied (the conjecture is that it is). It would also be nice to see some results where the assumptions are not satisfied: how does the algorithm perform then? That said, there are quite a few typos and confusing notation in the paper. It should be cleaned up. *** Section 2 l.89-90: (a) At one point you have $r_t(a_t)$ to be the reward, but shouldn't this depend on $s$? (b) Immediately afterwards, you write $r_t(\bar{s}_t, a_t)$ for the expected reward, but there is no expectation symbol. (c) You refer to $s$ as both a feature vector and a state vector. l. 90: $\theta$ is not explained. l.98-99: you introduce $\tilde{s}$, but I suppose it should be $\bar{s}$. l. 106-17, 112-113: I assume $\theta$ is some kind of parameter vector, please explain. *** Section 3 l. 162-176: Before $\bar{a}^*_t$ referred to the optimal action given that we are not playing action $0$. Please clarify what you mean by $\bar{a}$ and $\bar{a}_t$ *** References [1] https://arxiv.org/pdf/1107.4042.pdf [2] Bubeck and Cesa-Bianchi, "Regret Analysis of Stochastic and Nonstochastic Multi-armed Bandit Problems", Section 6.2. *** Post-author review I have read the authors' response and I agree, yet a comparison with those baselines would be good. In addition, seeing what happens when the assumptions are not satisfied would be nice.

Reviewer 2



In the context of multi-armed bandit problems for mobile health, modelling correctly the linearity or non-linearity of the rewards is crucial. The paper tries to take advantage of the fact that a general "Do nothing" action is generally available, that corresponds to the baseline. The idea is that, while the reward for each action may not be easily captured by a linear model, it is reasonable to assume that the relative error between playing an action and playing the "Do thing" action follows a linear model. The article revisits bandit strategies in this context. Comments: 1) Line 29: See also the algorithm OFUL, and more generally Kernel-UCB, GP-UCB, to cite a few other algorithms with provable regret minimization guarantees. 2) Line 35, 51, and so on: I humbly suggest you replace "mhealth" with something else, e.g. "M-health", since "mhealth" looks like a typo. 3) Question about the model: is \bar {f}_t assumed to be known? If not and if it is completely arbitrary, it seems counter-intuitive that one can prove any learning guarantee, given that a bandit setting is considered (only one action per step), and \bar {f}_t may change at any time to create maximal confusion. 4) Line 132: "might different for different" 5) Line 162: What is \bar{a}_t ? I can only see the definition for a_t and \bar{a}_t^\star. Likewise, what is \bar{a} on line 165? It seems these are defined only later in the paper, which is confusing. Please, fix this point. 6) Question: What is the intuition for choosing \pi_t to be (up to projection) P(s_{t,\bar a}^\top \bar \theta \geq 0) ? Especially, please comment on the value 0 here. 7) Lemma 5: Should 0 be replaced with a in the probability bound? 8) Section 9: I understand that you heavily used the proof of Agrawal and Goyal, but I discourage the use of sentences such as "For any filtration such that ... is true". I suggest you rather control probability of events. 9) Proof, line 62: there is a missing reference. ................. In the end the setting of the paper and definition of regret look a bit odd, but the authors provide an interesting idea to overcome the possibly nasty non-stationary term, thanks to randomization. Also deriving the weighted least-squares estimate and TS analysis in this context is interesting, even though relying heavily on previous existing analysis for TS. Finally, the idea to consider that only the relative errors follow a linear model might be generalized to other problems. Weak accept.